# Prevalence of Maternal Postpartum Depression, Health-Seeking Behavior and Out of Pocket Payment for Physical Illness and Cost Coping Mechanism of the Poor Families in Bangladesh: A Rural Community-Based Study

**DOI:** 10.3390/ijerph17134727

**Published:** 2020-07-01

**Authors:** Sheikh Jamal Hossain, Bharati Rani Roy, Aniqa Tasnim Hossain, Fardina Mehrin, SM Mulk Uddin Tipu, Fahmida Tofail, Shams El Arifeen, Thach Tran, Jane Fisher, Jena Hamadani

**Affiliations:** 1Maternal and Child Health Division (MCHD), International Centre for Diarrhoeal Disease Research, Bangladesh (icddr,b), 68 Shaheed Tajuddin Ahmed Sarani, Mohakhali, Dhaka 1212, Bangladesh; bharati.roy@icddrb.org (B.R.R.); aniqa.hossain@icddrb.org (A.T.H.); fardina.mehrin@icddrb.org (F.M.); mulk.tipu@icddrb.org (S.M.U.T.); ftofail@icddrb.org (F.T.); Shams@icddrb.org (S.E.A.); jena@icddrb.org (J.H.); 2Global and Women’s Health, School of Public Health and Preventive Medicine, Monash University, Melbourne, VIC 3004, Australia; Thach.Tran@monash.edu (T.T.); jane.fisher@monash.edu (J.F.)

**Keywords:** maternal depression, prevalence, health-seeking behavior, out of pocket payment, rural settings, Bangladesh

## Abstract

The burden of depression is high globally. Maternal depression affects the mother, the child, and other family members. We aimed to measure the prevalence of maternal postpartum depressive (PPD) symptoms having a child aged 6–16 months, health-seeking behavior for general illness of all family members, out of pocket (OOP) payments for health care and cost coping mechanisms. We conducted a cross sectional study with 591 poor families in rural Bangladesh. The survey was conducted between August and October, 2017. Information was collected on maternal depressive symptoms using the Self Reporting Questionnaire (SRQ-20), health-seeking behavior, and related costs using a structured, pretested questionnaire. The prevalence of depressive symptoms was 51.7%. Multiple logistic regression analysis showed that PPD symptoms were independently associated with maternal age (*p* = 0.044), family food insecurity (*p* < 0.001) and violence against women (*p* < 0.001). Most (60%) ill persons sought health care from informal health providers. Out of pocket (OOP) expenditure was significantly higher (*p* = 0.03) in the families of depressed mothers, who had to take loan or sell their valuables to cope with expenditures (*p* < 0.001). Our results suggest that postpartum depressive symptoms are prevalent in the poor rural mothers. Community-based interventions including prevention of violence and income generation activities for these economically disadvantaged mothers should be designed to address risk factors. Health financing options should also be explored for the mothers with depressive symptoms

## 1. Introduction

Depression is a disorder that affects people in multiple ways throughout their daily life. Historically, this has been overlooked or disregarded in lower and middle income countries (LMIC). The World Health Organization (WHO) reported that in the near future depression will shift from being the second most prominent cause of disease burden for women in high, middle and low income countries to the first place by the year 2030 [1]. Postpartum depression (PPD) is very common in low-income contexts. Previous studies conducted in rural Bangladesh report the prevalence of PPD between 18% and 52% among women [2,3,4,5].

Multiple factors affect maternal depression, such as a woman’s socioeconomic status, relationship with others, health and mental condition, but there are many other factors. The prominent factors reported for PPD in rural Bangladeshi women are poverty, insufficient nutrition, physical violence and domestic quarrels with a spouse and/or in laws, stress, any illness and previous mental disorders [2,3]. According to previous literature, young or adolescent mothers are more depressed than older mothers because they are often unprepared for the demands and responsibilities of parenting [6], and are also additionally at risk of having poor parenting skills. Several studies show that adolescent mothers have limited knowledge about child development, lack positive mother-infant interaction, have less capability to create a positive stimulating home environment, and are more likely to abuse their child [7,8,9].

Untreated maternal depression has detrimental effects on both mother and her child’s health and may even have a negative impact on the entire family life, and it can be the antecedent of chronic recurrent depression. This could also have an impact on emotional, behavioural and cognitive problems of their children later in life [10,11]. Depressed mothers may have a tendency of not being concerned with the benefit of exclusive breast feeding. Infant may suffer from extreme diarrhoea and malnutrition due to lack of exclusive breastfeeding. Inadequate mother-infant bonding can have negative impacts on child’s development and can lead to other infectious illnesses in many ways [12,13,14]. One study found that maternal PPD is associated with increased paternal depression and higher paternal stress due to parenting [15]. Another study documented that men who have low mood are more likely to abuse alcohol and behave violently and this then led to the maternal depression [16]. Indeed, maternal depression is often an indicator of insensitive or un-empathic family relationships and that the relationships of depressive symptoms to the outcomes are not as simple as stated here.

Although there has been consistent economic development in the last decade in Bangladesh, a recent study reported that 59% individuals visited either paraprofessional (those who have some formal training on allopathic treatment) or traditional or informal (no formal training on allopathic treatment) health care providers in rural Bangladesh [17]. Out of pocket (OOP) expenditure is the main source of health care financing in Bangladesh, and it is the highest in South Asia at 67% [18]. Excessive OOP expenditure may prompt financial catastrophe, distress financing, and impoverishment [19]. A study conducted in Bangladesh reported that 9% of households incurred catastrophic payments, 7% faced distress financing, and 6% experienced impoverishing health payments [20].

Proper health-seeking behaviour, awareness and knowledge about mental health or illnesses can reduce burden of illness and cost of treatment. Longitudinal relationship between pregnant mothers and health care providers from pregnancy to postpartum period was recommended [21]. According to WHO, those who are aware of seeking appropriate health care, when needed, can reduce child mortality due to acute respiratory infection by 20% [22]. So, understanding the burden of PPD and health-seeking behaviour and cost of health care of the family members for health problems where there was a depressed mother in the family can play an important role to formulate policy at primary level of health care. The objectives of this study were to assess the prevalence of postpartum depressive symptoms and treatment seeking behavior for health problems and out of pocket expenditure for illness and cost coping mechanism of the mothers and her other family members in rural Bangladesh.

## 2. Materials and Methods

### 2.1. Study Area

We conducted this study form August to October, 2017 in Ullapara sub district of rural Bangladesh. The health system of this area is almost akin to other sub-districts of the country. We conducted our study in 11 randomly selected unions, the lowest local government tier in Bangladesh, out of the existing 14. In Bangladesh, there are three of the old Wards in each Union, and these Wards are the territorial unit used by the rural health system, since much frontline health planning is done based on this geographic area (Ward). In the study area, more than seven thousand people reside in each Ward. We recruited around 18 mothers from each old Ward. The participant recruitment process and the inclusion and exclusion criteria are described elsewhere [23].

Our study area included the Upazilla health complex, the largest primary health care facilities in this subdistrict in Bangladesh which delivers health care services to more than five hundred thousand people, 10 Union Health and Family Welfare Centres, two Union Sub-Centres and 50 Community Clinics. The health system is pluralistic in the study area, with a number of private health care institutions and others health care providers, such as homeopathic doctors, medicine sellers, quacks and traditional healers.

### 2.2. Study Design

We collected this cross sectional data during collection of baseline information in a cluster randomised controlled trial.

### 2.3. Study Population and Recruitment

Information was collected from 591 participants from 33 randomised clusters for this study. The participants were the recipients or eligible to receive the maternity allowance programme from the Government of Bangladesh (GoB). The maternity allowance programme was designed for poor rural mothers as the social safety net programme of the Ministry of Women and Children Affairs (MoWCA), GoB. The mothers who had a child aged 6–16 months were selected for the study. The details of the participants have been discussed elsewhere [23].

### 2.4. Measurement

We used a self-reporting questionnaire (SRQ-20) to measure maternal depression status, since this tool has been used in other studies in Bangladesh [24,25,26], and an adopted cut point of 7 was chosen to define the presence of depression based on sensitivity 83%, specificity 80%, and range: 0–20 [27]. We measured Cronbach’s alpha of SRQ-20 of this study, and determining a value of 0.84. The questionnaire comprises 20 binary (yes = 1/no = 0) questions to measure the presence of depressive symptoms during past 30 days. Illness was defined as any family member perceived as being sick in the last month. We used a structured interviewer administered questionnaire for health-seeking behavior and related cost in the last month preceding the interview. Out of pocket expenditure was collected on consulting fee (user fee), transportation, diagnostic, medicine and accommodation cost. In order to ensure the reliability and validity of the tools, we developed the questionnaire based on a literature review. The research team reviewed the questionnaire. We piloted this questionnaire in a small sample (*n* = 30) of mothers living in Ullapara sub district and revised it accordingly to make it user friendly. This questionnaire was also previously used elsewhere in similar settings in Bangladesh [28].

Food insecurity was assessed using the Household Food Insecurity Access Scale (HFIAS). This questionnaire was developed by the Food and Nutrition Technical Assistance (FANTA) initiative of USAID and was used to measure household food security status. For this study, we generated a binary variable for household food security status: food secure and food insecure. We have previously used this questionnaire in our settings [29]. We used World Health Organization (WHO) Violence against women instruments to collect information on physical, mental and sexual violence [30]. We modified this instrument to use in our study. The tools were previously used in Bangladeshi settings [31]. Mothers and children’s height and weight were measured following WHO standards. We also collected socioeconomic and demographic information of the participants and the mothers were the respondents of this study.

### 2.5. Data Collection

Five data collectors with master degrees in psychology/child development/sociology collected the data. The respondents were the mothers. The data were collected through interview due to low literacy rates in the community. 

### 2.6. Ethics of the Study

This study was approved by the International Centre for Diarrhoeal Disease Research, Bangladesh (icddr,b) Institutional Review Board (Protocol Number: PR-17009). The participants provided written agreed consent to participate in the study.

### 2.7. Data Analysis

We recoded relevant information to develop different binary variables. For dependent variables- formal health professionals included registered doctors having Bachelor of Medicine and Surgery (MBBS) degree or Medical Assistant Training School (MATS) degree or government health care providers (nurses, community health care providers) and non-formal health care providers included all other professionals, i.e., traditional healers, village doctors, drug sellers and homeopathic doctors All factors, including need factors, enabling factors and predisposing factors that predict health care-seeking behavior were considered as independent variables. This was guided by an adaptation of Anderson’s behavioural model of health services and was previously used in Bangladesh to predict health-seeking behavior [32,33]. Housing index (person per living room) were constructed Anthropometry data were converted into length-for-age (LAZ), weight-for-length and weight-for-age (WAZ) z scores for children and body mass index-for-age z scores (BAZ) for mothers. The analyses involved descriptive statistics and logistic regression analysis to predict health-seeking behavior from formal health care providers. Independent t-test and Pearson chi-squared test were used to assess means and proportions to report sample characteristics for continuous and categorical variables, respectively. Mann Whitney U test was conducted for the OOP data. Finally, a logistic regression analysis was conducted to control for covariates. The data were analyzed using the IBM software Statistical Package for Social Science (SPSS) 20.0 for Windows (SPSS Inc., Chicago, IL, USA). The anthropometry measurements were analyzed using WHO Anthroplus software.

## 3. Results

The mothers’ mean (SD) age was 24.98 (3.11), with the range being 16–45 years. Around 14% mothers were adolescents. Most mothers in this population were somewhat literate, only 12% were illiterate, whereas 4% had more than 12 years of education. The majority of (96%) mothers were from the Muslim community, and 94% were housewives. The average Body Mass Index (BMI) of the mothers was 21.30 (3.42), and less than 2% were obese (BMI ≥ 30 kg/m^2^).

### 3.1. Prevalence of Depressive Symptoms

Table 1 shows socio-demographic characteristics of the respondents by depression status. Based on SRQ-20 (score > 7 out of 30), 51.7% of these poor women were depressed. Bivariate analysis showed that the women’s age (*p* = 0.004) and education (*p* = 0.015), and education of their husbands (*p* = 0.026), violence against mother and family food insecurity were significantly associated with depressive symptoms. Mothers’ depression status did not vary by child’s sex. 

Multiple logistic regression analysis showed that maternal PPD symptoms were independently associated with maternal age, family food insecurity and violence against the mothers (Table 2).

### 3.2. Distribution of Illness and Treatment Seeking Behaviour 

The mothers with depressive symptoms reported being sick in the last month, which was higher than the mothers without depressive symptoms (*p* = 0.004). When splitting the data by child’s sex, care seeking behavior for male child was similar in both depressed and non-depressed mothers (48% vs. 42%, *p* = 0.92). It also did not vary between depressed and non-depressed for mothers for female child (*p* = 0.28) and husband’s illness (*p* = 0.14). But the other family members of the mothers with depressive symptoms suffered significantly higher (*p* = 0.006) from illnesses than the family members of the mothers without depressive symptoms (Figure 1). 

Table 3 shows that in 83.6% of the families with a depressed mother at least one person was sick in the past month, compared to 74.6% of families of the non-depressed mothers (*p* = 0.008). Visits to health care providers for their treatment were made by 97.3% people with illness in the household where a depressed mother live, and it was 95.7% for people with illness in the household where there was no depressed mother, but the difference was not significant (*p* = 0.446). For treatment, 105 (42.3%) depressed mothers’ family members chose formal health care providers compared to 75 (37.3%) of non-depressed mothers’ family members, but the difference was not significant (*p* = 0.088). A majority of participants among the sample preferred to attend private facilities, with no significant difference (*p* = 0.359) between the depressed and non-depressed mother’s family members.

### 3.3. Health Care Expenditure 

Table 4 shows that in these populations, the median OOP for the health care was 300 Bangladeshi Taka (BDT) for the illness. The participants from depressed-mothers families spent a median of 320 BDT for their health, whereas those from non-depressed families spent a median of about 210 BDT in last one month for their health related problems. The OOP was significantly higher in the families where a depressed mother live compared to a non depressed mother’s family (*p* = 0.002)

Table 5 shows that more than 70% of families cope with the health care costs through payment by family members in these populations. To cope with OOP, 76% (*n* = 57) families with a depressed mother sold valuables or took a loan, compared to 24.0% [18] of the families with a non-depressed mother, and the difference was significant (*p* < 0.001). 

## 4. Discussion

This community-based study gave us an opportunity to measure the prevalence of depressive symptoms and its determinants in rural poor mothers having at least one child aged 6–16 months. We also aimed to generate a wider view and understanding of treatment-seeking behaviour, not only for the mother herself but for the children or anyone else in the family for their general health problems. In addition, we measured OOP and cost coping mechanism for the health care of the families with depressive symptoms. 

We found that maternal depressive symptoms were prevalent in this economically disadvantaged population. The family food insecurity and violence against women were the most important risk factors for developing depressive symptoms. We also found that the mothers with depressive symptoms and their other family members (except sons, daughters and husbands) suffered significantly more health problems than those of the non-depressed mothers. 

The members of the family as a whole received health care mostly from non-formal health care providers. OOP expenditure for health care was significantly higher in the families with a mother with depressive symptoms than the families without a mother with depressive symptoms. The coping strategies with health care cost were mostly through selling valuables or taking loans, thus burdening the family’s resources in the families of a mother with depressive symptoms. 

We documented that the prevalence of maternal depressive symptoms were 51.7% in this population having a child aged 6–16 months using SRQ-20 questionnaire. A community-based study conducted ten years before our study found depression prevalence of 52% among mothers having a child aged 6–12 months in rural Bangladesh using Center for Epidemiologic Studies Depression Scale (CESD) [5]. Although Bangladesh has seen considerable economic developed in the last decade, it seems that socioeconomic development did not influence the prevalence of depression. The slightly higher prevalence of depressive symptoms in our study could be due to participation of relatively poorer mothers in the study. A recent population-based study in three urban slums of Dhaka city, Bangladesh found a prevalence of 39.4% using the Edinburgh Postnatal Depression Scale (EPDS) among mothers within 12 months following delivery [34] and 46.2% in another urban settings using SRQ-20 [35]. In the South Asian context, the prevalence of depression varied in different countries. For example, in Pakistan, PPD was documented at 56% [36], in India it was found to be 22% in a systematic review [37], and in a hospital based survey in Nepal it was 30% [38]. A review of 203 studies documented the prevalence of PPD ranging from 1.9% to 82.1% in developing countries, where most of the studies used SRQ-20 scale [39]. However, settings e.g., economic status, sociodemographic, culture, inclusion and exclusion criteria of mothers, representativeness of the samples and psychometric properties of the assessment tools used in the studies varied widely. Since prevalence estimates are the results of all of these factors, it is difficult to compare these findings. A population based nationwide prevalence study could help understand the magnitude of the problems of the country.

A higher prevalence of depressive symptoms was found in our study to be supported by a recent community-based urban study in the country [34] and in other developing contexts, e.g., a rural study in Nigeria [40]. Higher numbers of child marriages, higher rates of girls’ dropout from high school despite GoB initiatives, little knowledge of the childcare of these mothers, increasing trends of nuclear families, and women’s participation in economic activities resulting in reduced time for child care might have negative impacts on a mothers’ higher mental stress, resulting in depression. In Bangladeshi culture, girls are brought up in a way that they should always prefer others in any circumstances. When these girls become mothers, their responsibilities are increased and they have to sacrifice their basic needs in many ways, which increases their mental stress. So, it can be speculated that the mothers feel helpless to care for their children, and this could help in developing depressive symptoms in those mothers. In our sample, higher depressive symptoms (Table 1) in lower educated mothers were comparable to a study in rural Mymensingh, Bangladesh [41] and another similar setting in Goa, India [42]. The difference between the two studies was that EPDS was used to measure depression at 2–3 months of postpartum in Mymensingh study and we used SRQ-20 among mothers having a child aged 6–16 months.

We found family food insecurity and violence against women were most important risk factors for developing depressive symptoms in the mothers (Table 2). An urban base study conducted in poor mothers in Dhaka, Bangladesh also documented that maternal mental health was associated with food insecurity [35]. Another rural based study in Bangladesh reported that violence against mothers during pregnancy was a risk factor for PPD symptoms at 6–8 months postpartum [3]. Intimate partner violence was also documented as a risk factor of mental disorder in the Vietnamese mothers [43]. This study only considered intimate partner violence, but we collected information on violence against women by any of the family members, and we found that husbands were the most frequent perpetrators.

A majority of families received health care from non-formal health care providers, irrespective of having a depressed mother in the family. This was an important finding of the study which gave us the clue that despite enormous efforts in the health sector and economic development of the country, large numbers of patients are still treated by non-formal health care providers. It is likely that economic constraints may be the reason for visits to non-formal health care providers by the participants of our study. Here it is worth mentioning that all of our participants were poor in this low resources setting. However, research findings showed that health care-seeking behavior is also largely dependent upon various socio-demographic determinants such as age, sex, family size, education, sanitation, and hygiene which can influence or negatively change health care seeking behavior [44].

A study to understand the pathway of health-seeking of the mental health patients in Bangladesh documented that 29% of such patients sought both the mental health and physical health care [45], which proved that mental health patients frequently visited health care providers for general illness. Nevertheless, the findings support that both non-formal and formal health care providers can play an important role to screen depression status of the mothers. If the non formal providers are also trained properly they can refer depressed mothers who were previously undiagnosed in the community to mental health care professional. Although the referral system is poor in Bangladesh [46], strengthening of the primary health care system for the mental health care referral system might improve screening and management of depression. So, it is possible for health care professionals to track and treat many depressed mothers, even at the level of primary health care. However, even in the presence of such facilities and referral systems, the access to care may still be hampered due to some demand barrier factors, such as stigma attached to mental health problems, lack of awareness, discrimination, and so on. So strategies need to be developed to combat social stigma, improve mass awareness, reduce discrimination of certain populations, and so on. 

OOP expenditure is significantly higher in families with a depressed mother. Only a small population of families spent more than 2000 BDT for their family’s medical care. In Bangladesh, 3.5% of families become poorer and fall under the poverty line due to out of pocket payments [47]. We found that cost-coping mechanisms such as selling valuable or receiving loan were two times higher in the family with a depressed mother compared to that of the families with a non-depressed mother, hence this cost may be catastrophic for many families. A study conducted in urban Bangladesh reported that 9% of households incurred catastrophic payments, 7% faced distress financing, and 6% experienced impoverishing health payments. This study further recorded that the households spent about 11.0% of their total household budget on health care [20]. It is important to quantify the economic impact of depression on families in order to understand the economic burden of depression at a national level. Another study documented that to be poor is highly stressful and one of the major contributors to low mood [48].

This is the first paper that documents the prevalence of postpartum depressive symptoms in a very poor population who are under a safety net programme (maternal allowance) or eligible to receive maternal allowance from the Bangladesh government in a community-based rural setting. This study focused on health-seeking behavior of the family members (any family member, including the mother) and their out of pocket payments and cost coping mechanisms for health care, rather than focusing only the depressed mothers. The questionnaires used in this study were all previously validated and used in Bangladesh. Therefore, the findings can be considered valid. The information provided by this study highlights the importance of proper health-seeking behaviour to reduce costs, especially for the families in which there was a mother with depressive symptoms. 

The cross sectional nature of the study, although it was a community-based study, the design and the limited variables considered for the analysis may not show the real picture in these families, which may limit the generalizability of the outcomes. We considered direct cost only to measure the cost of health care but indirect cost measurement would have had more implications in this study.

## 5. Conclusions

Postpartum depressive symptoms are prevalent in the poor rural mothers. Community-based interventions including prevention of violence and income generation activities for these economically disadvantaged mothers should design to address risk factors. Health financing options should also be explored for mothers with depressive symptoms.

## Figures and Tables

**Figure 1 ijerph-17-04727-f001:**
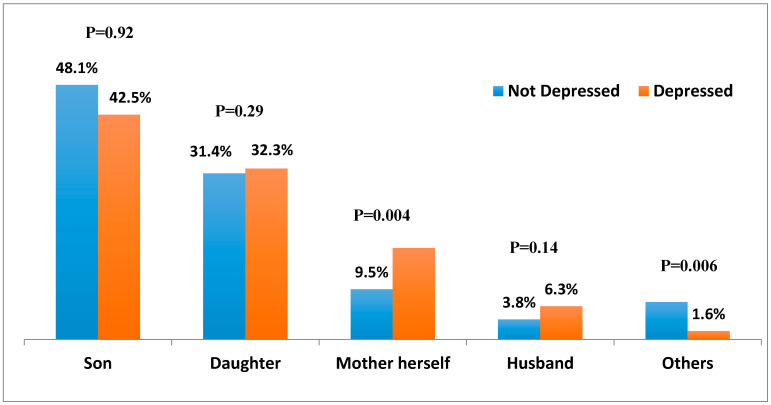
Distribution of illness in the family members (mother herself or offspring or husband or anyone else) in the last one month by depression status (*n* = 464).

**Table 1 ijerph-17-04727-t001:** Socio-demographic characteristics by depression status of the respondents.

Characteristics	Not Depressed		Depressed		*p* Value
	*n* = 284	Percentage/Mean (95% CI)	*n* = 307	Percentage/Mean (95% CI)	
**Children Sex**					
Male child	144	50.7%	161	52.4%	0.367
Female child	140	49.3%	146	47.6%	
**Household Size**	284	5.29 (5.05–5.53)	307	5.00 (4.82–5.18)	0.06
**Mother’s age** (years)	284	24.4 (23.8–25.0)	307	25.6 (25.0–26.1)	0.007
**Mother’s Education ***					
Low education (0–5 years)	132	46.5%	171	55.7%	0.015
Higher education (>5 years)	152	53.5%	136	44.3%	
**Father’s Education ***					
Low education (0–5 years)	158	55.6%	196	59.9%	0.026
Higher education (>5 years)	126	44.4%	111	40.1%	
**Having Son**					
No	91	32.0%	87	28.3%	0.187
Yes	193	68.0%	220	71.7%	
**Monthly Family Income**					
Low income (≤5300 BDT)	46	16.2%	53	17.3%	0.407
High income (>5300 BDT)	238	83.8%	254	82.7%	
**Housing Index (Type) ****					
Well built house	19	6.7%	18	5.9%	0.403
Not well built house	265	93.3%	289	94.1%	
**Mothers BMI** ^#^					
Normal (BMI: 18.5–24.0)	185	65.1%	198	64.5%	0.911
Under weight (BMI < 18.5)	60	21.1%	69	22.5%	
Over weight (BMI > 24.0)	39	13.8%	40	13.0%	
**Children Stunting**					
Not-Stunted (LAZ ≥ −2SD)	228	80.3%	247	80.5%	0.520
Stunted (HAZ < −2SD)	56	19.7%	60	19.5%	
**Children Wasted**					
Not-Wasted (WAZ ≥ −2SD)	243	85.6%	261	85.0%	0.472
Wasted (WAZ < −2SD)	41	14.4%	46	15.0%	
**Loan Status**					
Do not have loan	126	44.5%	128	41.7%	0.271
Has loan	157	55.5%	179	58.3%	
**Crowding index ^##^**	284	3.15 (3.01–3.30)	307	3.31 (3.16–3.46)	0.140
**Violence**					
No violence	255	89.8%	208	67.8%	<0.001
Violence	29	10.2%	99	32.2%	
**Household Food Security Status**					
Secured	185	65.1%	112	36.5%	<0.001
Not secured	99	34.9%	195	63.5%	
**Ill Person**					
Daughter	66	23.2%	83	27.0%	0.526
All other	117	41.2%	116	37.8%	
Son	101	35.6%	108	35.2%	

*t*-test for continuous variable, Chi-square test for categorical variable; * Low education: 0–5 years of education level, Higher education: >5 years of education level; **^#^** Normal BMI: 18.5–24.0, Underweight BMI < 18.5, Overweight BMI > 24.0; ** A housing index was created based on the condition of the floor, roof and walls of the house (well built means better); ^##^ Crowding index was created through number of people divided by number of bed room in a household.

**Table 2 ijerph-17-04727-t002:** Determinants of maternal depressive symptoms within 14 months of postpartum.

Characteristics	OR (95% CI)	*p*-Value
**Children sex**		
Male	Ref	
Female	0.87 (0.56, 1.37)	0.549
**Family Size (person)**	0.93 (0.84, 1.03)	0.149
**Woman’s age (years)**	1.04 (1.00, 1.08)	0.044
**Woman’s education ***		
Low education (0–5 years)	Ref	
High education (>5 years)	0.99 (0.67, 1.46)	0.955
**Husband’s education ***		
Low education (0–5 years)	Ref	
High education (>5 years)	0.95 (0.65, 1.41)	0.812
**Ill person**		
Daughter	Ref	
Son	0.79 (0.44, 1.41)	0.420
All other (other than daughter and son)	0.80 (0.49, 1.31)	0.367
**Household food security status**		
Secured	Ref	
Not secured	2.69 (1.88, 3.84)	<0.001
**Violence against woman**		
No violence	Ref	
Violence	3.52 (2.19, 5.66)	<0.001
**Mothers BMI ^#^**		
Normal (BMI:18.5–24.0)	Ref	
Under weight (BMI < 18.5)	1.05 (0.67, 1.65)	0.821
Over weight (BMI >24.0)	0.95 (0.56, 1.63)	0.864

* Low education: 0–5 years of education level, Higher education: >5 years of education level; ^#^ Normal BMI: 18.5–24.0, Underweight BMI < 18.5, Overweight BMI > 24.0.

**Table 3 ijerph-17-04727-t003:** Illness, visit of health care provider and choice of health care provider and facility of the family member (any family member, including the mothers) in the last one month by depression status.

Status		Not Depressed *n* (%)	Depressed *n* (%)	*p* Value
Illness (*n* = 588)	Yes	211 (74.6)	255 (83.6)	0.008
	No	72 (25.4)	50 (16.4)	
Health care provider visited (*n* = 465)	Yes	201 (95.7)	248 (97.3)	0.446
	No	9 (4.3)	7(2.7)	
Choice of health care provider (*n* = 449)	Formal	75 (37.3)	105 (42.3)	0.088
	Informal	126 (62.7)	143 (57.7)	
Preference of health facilities (*n* = 446)	Government facilities	29 (40.8)	42 (59.2)	0.359
	Private facilities	47 (40.9)	68 (59.1)	
	Others	124 (47.7)	136 (52.3)	

**Table 4 ijerph-17-04727-t004:** Distribution of out of pocket (OOP) payment (mean, SD) of the families in last one month by depression status (*n* = 449).

Out of Pocket Payment (OOP) *	Not Depressed (Seek Health Care)	Depressed (Seek Health Care)	*p*-Value ^#^
Consulting Fee	81.79 (339.5)	113.84 (269.4)	0.010
Transport	40.39 (170.5)	96.50 (706.9)	0.052
Diagnostic	68.23 (368.5)	87.15 (418.4)	0.162
Medicine	393.48 (1069.8)	629.96 (1617.0)	0.003
Accommodation & others	19.81 (297.4)	13.73 (110.7)	0.082
Total	608.0 (1670.7)	974.7 (2419.1)	0.002

^#^ Mann whitney U test; * OOP cost was in BDT (Bangladeshi Taka).

**Table 5 ijerph-17-04727-t005:** Cost coping mechanism of the families for health care in the last month by depression status (*n* = 449).

Cost Coping Mechanism	Not Depressed *n* (%)	Depressed *n* (%)	*p* Value ^#^
Own family member/relative	164 (48.2)	176 (51.8)	0.918
Loan/selling valuables	18 (24.0)	57 (76.0)	<0.001
Other sources	19 (55.9)	15 (44.1)	0.347

^#^ Chi square test.

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
