# Peer review of "Prevalence of Maternal Postpartum Depression, Health-Seeking Behavior and Out of Pocket Payment for Physical Illness and Cost Coping Mechanism of the Poor Families in Bangladesh: A Rural Community-Based Study"

_ijerph, 2020, doi:10.3390/ijerph17134727_

Round 1
Reviewer 1 Report
The article is very interesting and deals with a very salient topic: postpartum depression, a topic particularly relevant in the context studied.
The bibliography of the introductory part is pertinent, however an important aspect is missing related to the relationship with healthcare facilities and the relationship of obstetricians and gynecologists with women. In this regard, for any further information, please note
Rania (2019). Giving voice to my childbirth experiences and making peace with the birth event: the effects of the first childbirth on the second pregnancy and childbirth. 10.1177/2055102919844492
As for discussion and conclusions, the authors are invited to investigate the limits of the study, any future developments of the research but above all to expand the theoretical and practical implications that can be drawn from the data obtained
Reviewer 2 Report
This study focuses on the prevalence of maternal postpartum depressive symptoms among poor families in rural Bangladesh. Its outcomes are rightfully claimed to be of importance for the policy formulation in regard to mental health action plan in low resource settings.
Well done, indeed. I really liked Your valuable manuscript (MS). My sincere congrats to all authors.
Based on its scientific merit, I can most kindly recommend Your highly informed (knowledgeable) MS for publishing in a forthcoming issue of International Journal of Environmental Research and Public Health (IJERPH), MDPI.
Overall Recommendation: Accept after minor revision
If You are in position to act in such a way, please, kindly improve a bit the English language and style.
At least in my humble opinion, this MS has a real potential to be clearly recognised and appreciated by the global academic/research community, once when published. If English polished, it might earn a number of hetero-citations (=be frequently cited in terms of its hetero-citations), after being launched.
Last but not least, very best of (research) luck ahead to You all.
Reviewer 3 Report
Please see my comments below:
1) the manuscript was not properly formatted and typo errors can be seen throughout the manuscript.
2) Introduction: the authors did not introduce any information on the topic from Bangladesh. This is very important because this will set the stage for the manuscript. Please revise significantly.
3) the methods were unorganised and need to be significantly improved. For example, when was the study conducted? What are the inclusion and exclusion criteria? What are the parameters measured? How was the questionnaire developed? Cronbach’s alpha value? All these are needed to be reported.
4) Figure 1: please add p-value for the comparison to indicate significance.
5) Results: they are not presented adequately. How about the descriptive statistics? The participant characteristics? This is not reported.
6) Discussion: It does not flow well. What are the major findings from the study? What are the study limitations?
7) How about other studies when compared with the authors’ findings? Why is it important to conduct such a study?
8) Explain the study limitations and strengths more details.
9)Conclusion: This does not tell us anything about the study. Please revise.
10) references: this is not properly formatted. Please revise. This shows that the authors do not pay attention to details and need to be improved.
Reviewer 4 Report
This manuscript does not provide useful new information. It has significant limitations with methodology as well as interpretation of the results. The findings are not novel or compelling.
Major Comments:
- Define the basis of participant’s recruitment.
- Give details of the questionnaire.
- Define Food Insecurity Access Scale (HFIAS).
- Out of pocket expenditure varies from person to person. How the data is equalized?
- Define the housing index.
- Define low and high education.
- Define well/not well build house.
- Add units in every table.
Round 2
Reviewer 3 Report
The authors have answered all my queries except, except Cronbach’s alpha value. Please perform a statistical test to get this values. Although the questionnaires were used by the same research team in previous studies in our settings, the value is still needed to be measured.
Author Response
Comments and Suggestions for Authors
The authors have answered all my queries except, except Cronbach’s alpha value. Please perform a statistical test to get this values. Although the questionnaires were used by the same research team in previous studies in our settings, the value is still needed to be measured.
Response: Thanks for your query. The Cronbach's alpha for SRQ-20 of this study was 0.84. We have added this in the manuscript in the measurement section in page#3.
Reviewer 4 Report
In the revised paper, the authors have more explained the methods and results of their study and have added some more useful data. The paper is now acceptable for publication.
Author Response
Comments and Suggestions for Authors
In the revised paper, the authors have more explained the methods and results of their study and have added some more useful data. The paper is now acceptable for publication.
Response: Thanks for your comments and review.